# Exploiting volumetric wave correlation for enhanced depth imaging in scattering medium

Ye-Ryoung Lee [1,2,3,4,7], Dong-Young Kim[1,2,7], Yonghyeon Jo [1,2], Moonseok Kim[5,6] & Wonshik Choi [1,2] ✉

Imaging an object embedded within a scattering medium requires the correction of complex sample-induced wave distortions. Existing approaches have been designed to resolve them by optimizing signal waves recorded in each 2D image. Here, we present a volumetric image reconstruction framework that merges two fundamental degrees of freedom, the wavelength and propagation angles of light waves, based on the object momentum conservation principle. On this basis, we propose methods for exploiting the correlation of signal waves from volumetric images to better cope with multiple scattering. By constructing experimental systems scanning both wavelength and illumination angle of the light source, we demonstrated a 32-fold increase in the use of signal waves compared with that of existing 2D-based approaches and achieved ultrahigh volumetric resolution (lateral resolution: 0.41 $\mu m$, axial resolution: 0.60 $\mu m$) even within complex scattering medium owing to the optimal coherent use of the broad spectral bandwidth (225 nm).

Complex media such as brain tissues under a skull induces both multiple scattering and sample-induced phase retardations, which make it difficult to extract single-scattered waves carrying the object information for high-resolution imaging. The single-scattered component of each planar wave constituting a focused illumination is attenuated exponentially with depth due to multiple scattering. The phase retardations further attenuate the peak intensity by hampering the constructive interference of the single-scattered waves in forming a focused spot. It has been critical to find and compensate for the phase retardations for recovering the optimal resolving power in deep imaging[1–3]. While there have been technically diverse approaches[4–12], their strategies have common ground. Essentially, these methods control phase retardation, either physically or computationally, to optimize image formation by the signal waves in each 2D image. For instance, the sharpness and brightness of the reconstructed image are the most widely used image metrics. The fidelity of distortion compensation is mainly determined by the relative magnitude of signal waves to multiple scattering backgrounds induced by wave deflection. Therefore, it is necessary to exploit more signal waves than those contained in 2D images to enhance the imaging depth beyond that of the existing approaches. A straightforward extension is the use of the wave correlation of a volumetric image, instead of a 2D image. While this is likely to increase the imaging depth according to the number of depth sections contained in the volume, coherent volumetric imaging is a prerequisite.

Ideal volumetric imaging requires the coverage of the two fundamental degrees of freedom: the propagation angle and wavelength. Wide angular coverage enables confocal depth sectioning, broad spectral coverage enables temporal (or coherence) depth gating, and their combination ensures maximum depth sectioning for volumetric imaging. Most optical imaging modalities have chosen to cover a wide range of illumination angles at once, instead of scanning the angles one

[1]Center for Molecular Spectroscopy and Dynamics, Institute for Basic Science, Seoul 02841, Korea. [2]Department of Physics, Korea University, Seoul 02841, Korea. [3]Institute of Basic Science, Korea University, Seoul 02841, Korea. [4]Department of Physics, Konkuk University, Seoul 05029, Korea. [5]Department of Medical Life Sciences, College of Medicine, The Catholic University of Korea, Seoul 06591, Korea. [6]Department of Biomedicine & Health Sciences, College of Medicine, The Catholic University of Korea, Seoul 06591, Korea. [7]These authors contributed equally: Ye-Ryoung Lee, Dong-Young Kim. ✉e-mail: wonshik@korea.ac.kr

by one. Examples are focused illumination in confocal imaging[13] and spatially incoherent illumination[14]. Likewise, a broadband light source containing multiple wavelengths is illuminated at once to gain depth resolution in time-domain optical coherence tomography (OCT)[15]. Optical coherence microscopy covers a broad range of illumination angles and wavelengths altogether to maximize both lateral and axial resolutions[16–19]. While the simultaneous wideband coverage ensures fast imaging speed and a simple experimental setup, these methods have critical deficiencies in deep-tissue imaging. Wave retardation mentioned above consists of angle-dependent phase retardation called aberration and wavelength-dependent phase retardation termed dispersion, which we term spectro-angular dispersion. The integral detection of mixed illumination angles and wavelengths makes it difficult to individually compensate for spectro-angular dispersion. Therefore, the reduction in resolving power has been inevitable due to the broadening of the focus and temporal gating window when it comes to deep optical imaging.

To partially resolve these issues, there have been attempts to scan either the wavelength or illumination angle. Spectral-domain OCT measures the response to individual wavelengths while either using a focused illumination for angular coverage[18,20–22] or planar wave illumination without angular coverage[23]. This method can correct spectral dispersion, but the aberration correction is either out of reach or incomplete. In another noteworthy approach, wide-field complex-field maps are recorded to scan either the illumination angle[10,11] or the position of the point illumination[12,24] of a broadband light source, which makes it possible to correct complex aberrations such as those from an intact mouse skull[12]. However, the presented approaches have difficulty in addressing spectral dispersion.

In our study, we aim to image an object embedded within a complex medium that induces multiple scattering and spectro-angular dispersions. We intend to reconstruct the object image with an ideal diffraction-limited resolution by separating the single-scattered waves from multiple scattering backgrounds and correcting the spectro-angular dispersions that the single-scattered waves have experienced. To this end, we propose volumetric reflection-matrix microscopy (VRM) that employs a wavelength-tunable laser and a galvanometer mirror for scanning both the wavelength and illumination angle. We record the complex-field maps of backscattered waves for individual combinations of wavelengths and illumination angles. These allow us to construct a coherent volumetric reflection matrix that contains all the signal waves from the covered volume. The spectral bandwidth coverage is 225 nm ranging from 535 nm to 760 nm, and the angular coverage is up to a numerical aperture (NA) of 1.0. A volumetric image cannot be obtained by the simple addition of acquired images. Therefore, we develop a general framework that universally combines both the wavelength and angle for the reconstruction of volumetric images based on the object momentum conservation principle. On this basis, we develop an algorithm to correct spectro-angular dispersion and achieve ultrahigh volumetric resolution (lateral resolution: 0.41 $\mu m$, axial resolution: 0.60 $\mu m$) even within a complex medium whose effective optical thickness ($l_{eff}$) is 9.4 $l_s$ with the scattering mean free path $l_s = 56.1$ μm and 83.1 μm for the wavelength of 535 nm and 760 nm, respectively.

With the coherent volumetric information at hand, we further develop image reconstruction methods that exploit signal waves from the volume. Specifically, we propose two unique approaches taking the advantage of the volumetric reflection matrix—progressive depth imaging, and volumetric dispersion correction. In the progressive depth imaging, dispersions at shallower depth are used to resolve those at a deeper depth. By applying the dispersion correction of a relatively shallow depth ($l_{eff} \sim 6l_s$) for imaging a deeper depth ($l_{eff} \sim 9l_s$), a 32-fold increase in the effective single-scattering intensity is achieved. In volumetric dispersion correction, the wave correlation of single scattering signals from a volume, not a 2D plane, is used to cope with multiple scattering backgrounds. We demonstrate a 5.6-fold enhancement of the effective detection channels by exploiting the volumetric reflection matrix from eight different depths. These methods enable us to reach an unprecedented imaging depth where multiple scattering and dispersion are so strong that a target cannot be reconstructed with the conventional 2D-based approaches.

## Results

The results section consists of five subsections: acquisition of a volumetric reflection matrix, a general framework of volumetric image reconstruction, correction of spectro-angular dispersion for optimal volumetric spatial resolution, progressive depth imaging, and volumetric dispersion correction. In the first subsection, we lay out the experimental procedure of taking a volumetric reflection matrix. In the second and third subsections, we introduce the volumetric image reconstruction framework embracing both propagation angles and wavelengths and its extension to find and correct the spectro-angular dispersions induced by the scattering medium in the presence of substantial multiple scattering backgrounds. In the fourth and fifth subsections, we introduce methods to exploit the developed volumetric reflection matrix imaging framework for increasing the achievable imaging depth beyond 2D-imaging-based approaches.

### Effects of multiple scattering and spectro-angular dispersion on microscopic image formation

To obtain a high-resolution image of a target object embedded within a complex medium, it is necessary to extract single-scattered waves that have been scattered only once by the target object. However, complex media such as brain tissue under a skull induces both multiple scattering and spectro-angular dispersions, as illustrated in Fig. 1a. Multiple scattering and spectro-angular dispersions make it difficult to extract single-scattered waves carrying the object information. The single-scattered component of each planar wave constituting a focused illumination is attenuated by a factor of $e^{-2z/l_s}$ at the depth $z$ due to multiple scattering (Fig. 1b). The spectro-angular dispersions further attenuate the peak PSF intensity by a factor of $\eta^2$ by hampering the constructive interference of the single-scattered waves in forming a focused spot, where $\eta$ is the Strehl ratio (Fig. 1c). Therefore, the single-scattered waves making the roundtrip to the target object are attenuated by $\eta^2 e^{-2z/l_s}$, making them obscured by the multiple scattering. To resolve the attenuation factor $\eta^2$, we present a method to find and correct the spectro-angular dispersions $\phi_{in}(\mathbf{k}_{in}, \lambda)$ and $\phi_o(\mathbf{k}, \lambda)$ in the input and output pathways, respectively, by considering individual wavelengths and propagation angles of light waves rather than the integral coverage of either wavelength or angle (Fig. 1a). For filtering out the multiple-scattered waves ($E_M$), we exploit confocal and coherence gating.

### Acquisition of a volumetric reflection matrix

For obtaining a volumetric reflection matrix, the electric field of the reflected wave, $E(\mathbf{r}; \boldsymbol{\theta}_{in}, \lambda)$, was measured for each combination of incident angle ($\boldsymbol{\theta}_{in}$) and wavelength ($\lambda$). Fig. 2a shows the experimental setup.(see Supplementary Information Section I for the details about the experimental setup and the distinction of our setup from existing modalities). A supercontinuum laser (NKT Photonics, model EXR-15) was used to scan the wavelength of the light source. The wavelength $\lambda$ was scanned from 542.5 to 752.5 nm at 7.5 nm intervals with the number of sampled wavelengths $N_\lambda = 29$. Considering that the bandwidth of the light source at each wavelength was $\Delta\lambda_S = 15$ nm, the full spectral bandwidth was $\Delta\lambda_F = 225$ nm. The expected depth resolution set by $\Delta\lambda_F$ was -0.59 μm at the sample (refractive index: 1.4). The coherence length set by $\Delta\lambda_S$ (8.88 μm at the sample) determines the

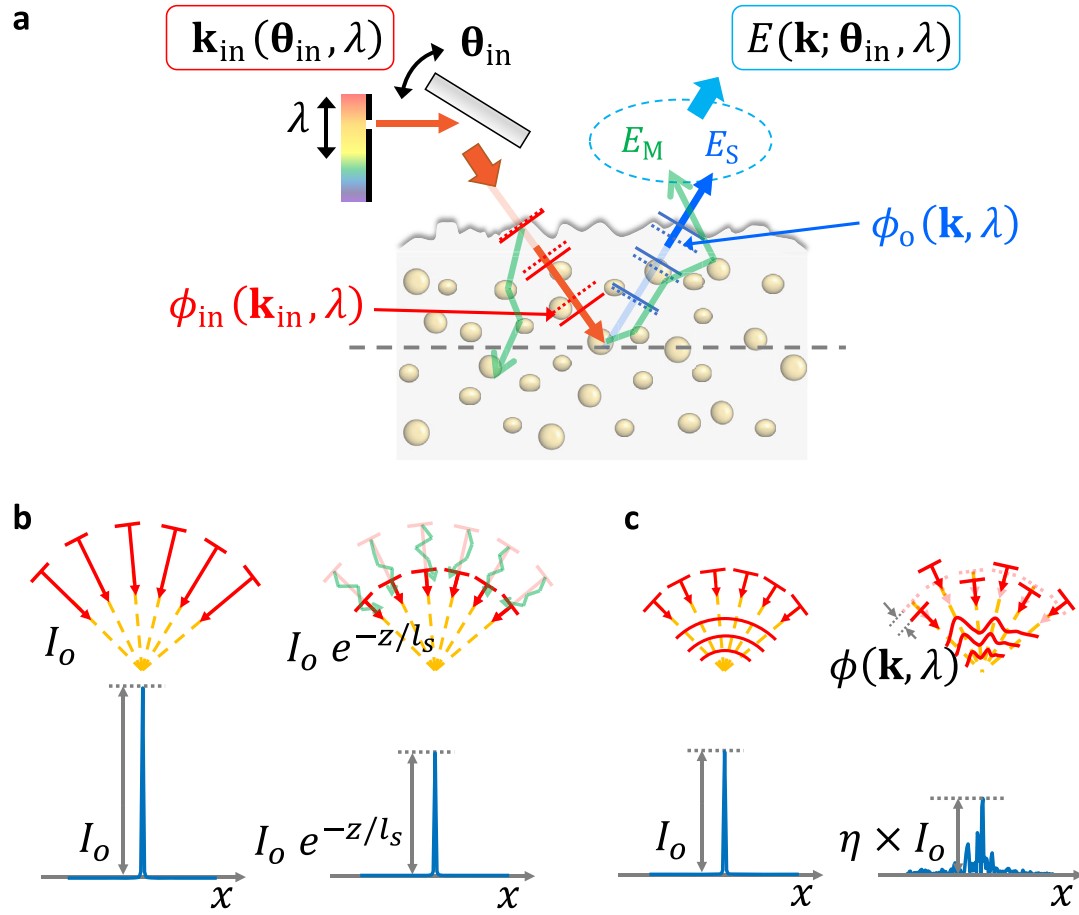

**Fig. 1 | Effects of multiple scattering and spectro-angular dispersions on image formation. a** For an incident planar wave with incidence angle $\theta_{in}$ and wavelength $\lambda$, its small fraction termed a ballistic wave indicated by a red arrow propagates straight through the scattering medium while maintaining its incident wavevector set by $\mathbf{k}_{in}(\theta_{in}, \lambda) = \frac{2\pi}{\lambda}\sin\theta_{in}$. The rest indicated by a green arrow contribute to multiple-scattered waves. The incident ballistic wave is scattered by an object on the sample plane to a wavevector $\mathbf{k}$, and its small fraction travels ballistically to the detector as indicated by a blue arrow, which we term a single-scattered wave $E_S$. All the multiple-scattered waves generated during the roundtrip contribute to the multiple-scattered wave $E_M$ at the detector. The single-scattered wave experiences the spectro-angular dispersion $\phi_{in}(\mathbf{k}_{in}, \lambda)$ in the input pathway and $\phi_o(\mathbf{k}, \lambda)$ in the output pathway, which jointly hampers microscopic image formation. **b** Effect of

multiple scattering on the image formation. The single-scattered component (red arrows) of each planar wave constituting a focused illumination is attenuated by $\exp(-z/l_s)$ with $l_s$ the scattering mean free path, and multiple-scattered waves are generated as a consequence (green arrows). Peak intensity of the PSF shown with the blue is attenuated by the same factor. The same attenuation occurs in the return path such that the net attenuation is $\exp(-2z/l_s)$. **c** Effect of the spectro-angular dispersion on the image formation. Angle- and wavelength-dependent phase retardations ($\phi(\mathbf{k}, \lambda)$) by the complex medium further attenuate the peak intensity of the PSF by hampering the constructive interference of the single-scattered waves in forming a focus. Strehl ratio $\eta$ is defined by the attenuation of the peak PSF intensity by the spectro-angular dispersions. The same attenuation occurs in the return path such that the net attenuation is $\eta^2$.

volumetric depth coverage. The incident angle at the sample was scanned with a galvanometer mirror to cover uniformly the NA of the objective lens (60 x Nikon CFI Apochromat, 1.0 NA, 2.8 mm WD). The number of scanned incident angles was 2400. The electric field of the backscattered wave $E(\mathbf{r}; \theta_{in}, \lambda)$ was measured via off-axis holographic phase microscopy.

We prepared a scattering medium composed of randomly dispersed $TiO_2$ particles with a 1 μm mean diameter in polydimethylsiloxane (PDMS) to obtain a representative volumetric sample. The focus of the objective lens was fixed at the center of the volumetric object (indicated by the gray dashed line in Fig. 1a) to minimize the loss of image information from defocused depths. We then obtained a set of reflected waves, $E(\mathbf{r}; \theta_{in}, \lambda)$, for all the combinations of ($\theta_{in}, \lambda$) (Fig. 2b). Subsequently, we determined the spatial frequency spectrum of the reflected wave $\widetilde{E}(\mathbf{k}; \theta_{in}, \lambda)$ by taking the Fourier transform of $E(\mathbf{r}; \theta_{in}, \lambda)$ with respect to $\mathbf{r}$. Since the object function is determined by the momentum difference $\mathbf{k} - \mathbf{k}_{in}$, where

$\mathbf{k}_{in}(\theta_{in}, \lambda) = \frac{2\pi}{\lambda}\sin\theta_{in}$, we converted the basis from $\theta_{in}$ to $\mathbf{k}_{in}$ for all the recorded images (Fig. 2c). The measured output fields of the same $\theta_{in}$ and different $\lambda$ (green box in Fig. 2b) end up having different $\mathbf{k}_{in}$ (green box in Fig. 2c). Both $\mathbf{k}_{in}$ and $\mathbf{k}$ have finite bandwidths ($|\mathbf{k}_{in}|, |\mathbf{k}| \leq 2\pi\alpha/\lambda$), which is determined by the numerical aperture $\alpha$ of the objective lens.

## A general framework of volumetric image reconstruction

In this section, we describe the volumetric image reconstruction from the recorded volumetric reflection matrix $\widetilde{E}(\mathbf{k}; \mathbf{k}_{in}, \lambda)$ based on object momentum conservation principle. We consider the cases when there exist only multiple scattering backgrounds with no spectral and angular dispersions. Each of the measured reflected waves consists of single- and multiple-scattered waves from the volumetric object: $\widetilde{E}(\mathbf{k}; \mathbf{k}_{in}, \lambda) = \widetilde{E}_S(\mathbf{k}; \mathbf{k}_{in}, \lambda) + \widetilde{E}_M(\mathbf{k}; \mathbf{k}_{in}, \lambda)$. The measured single-scattered wave is the superposition of those reflected from various depths of the volumetric object (see Supplementary Information

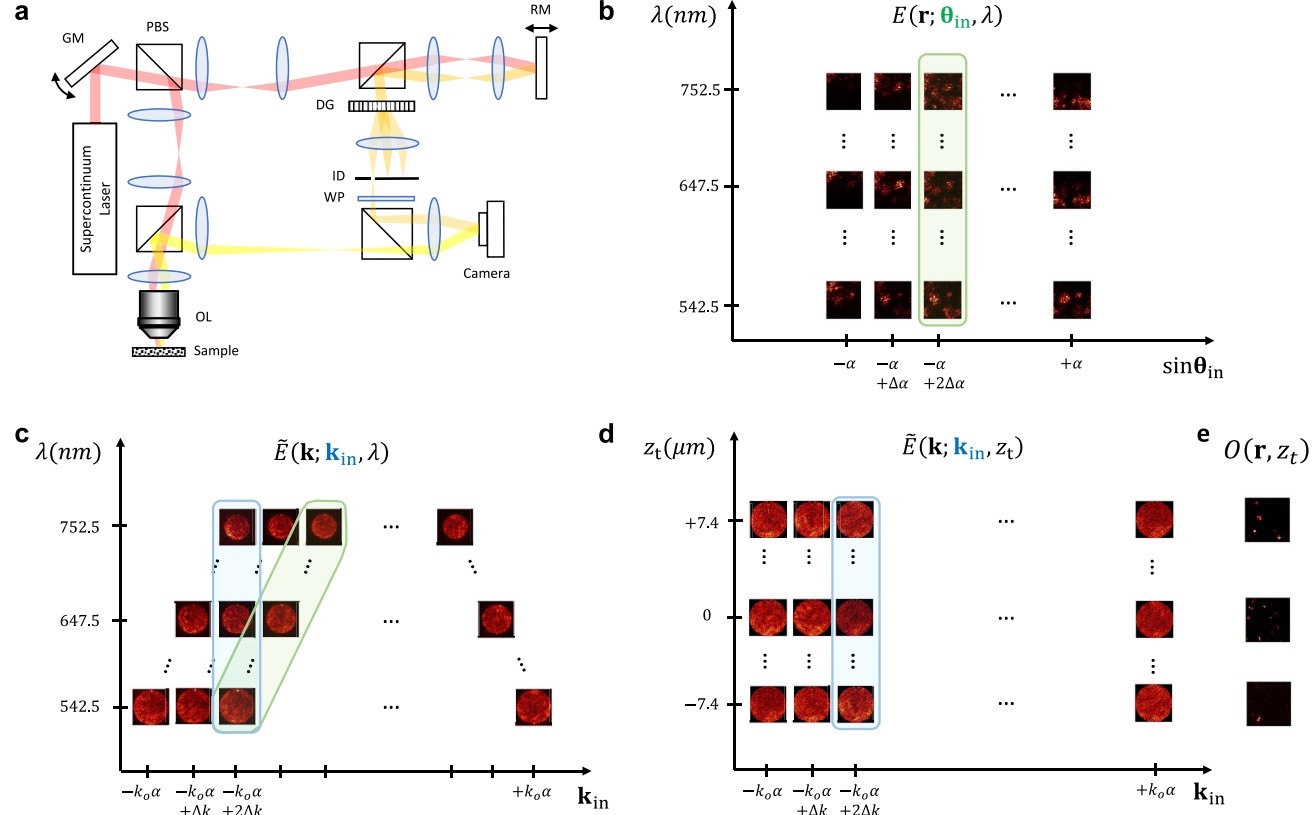

**Fig. 2 | Spectro-angular representation of volumetric image reconstruction framework. a** Experimental setup. Reflected wave $E(\mathbf{r}; \boldsymbol{\theta}_{in}, \lambda)$ was measured by scanning both incident angle $\boldsymbol{\theta}_{in}$ and wavelength $\lambda$ via off-axis holographic phase microscopy. GM: a galvanometer scanning mirror, PBS: a polarizing beam splitter, OL: the objective lens, RM: the reference mirror, DG: the diffraction grating, ID: the iris diaphragm, WP: A wave plate. **b** A measured set of $E(\mathbf{r}; \boldsymbol{\theta}_{in}, \lambda)$ displayed with respect to $\lambda$ and $\boldsymbol{\theta}_{in}$. Each image has view field of $22 \times 22 \mu m^2$. **c** Spatial frequency

spectrum of reflected wave $\widetilde{E}(\mathbf{k}; \mathbf{k}_{in}, \lambda)$ obtained by taking Fourier transform of $E(\mathbf{r}; \boldsymbol{\theta}_{in}, \lambda)$ with respect to $\mathbf{r}$. They are displayed with respect to $\lambda$ and $\mathbf{k}_{in}$. **d** A set of $\widetilde{E}_{cg}(\mathbf{k}, z_t; \mathbf{k}_{in})$ obtained by the summation of $\widetilde{E}(\mathbf{k}; \mathbf{k}_{in}, \lambda)$ with respect to $\lambda$ after input/output defocus compensation to $z_t$. **e** Coherence-gated confocal image of object, $O(\mathbf{r}, z_t)$, obtained by taking inverse Fourier transform of confocal and coherence-gated field $\widetilde{E}_{ccg}(\mathbf{K}, z_t)$, which is the summation of spectral shift-compensated $\widetilde{E}_{cg}(\mathbf{K} + \mathbf{k}_{in}, z_t; \mathbf{k}_{in})$ with respect to $\mathbf{k}_{in}$, where $\mathbf{k} = \mathbf{K} + \mathbf{k}_{in}$.

Section II for the detailed description):

$$\widetilde{E}_S(\mathbf{k}; \mathbf{k}_{in}, \lambda) = \int e^{-2z/l_s} \widetilde{O}(\mathbf{k} - \mathbf{k}_{in}, z) e^{ik_z^{in}(\lambda)z} e^{ik_z(\lambda)z} dz. \quad (1)$$

Here, $z$ is the depth of a volumetric object with respect to the objective focus, and $\widetilde{O}(\mathbf{k}, z)$ is the spatial frequency spectrum of the object function $O(\mathbf{r}, z)$ at $z$. The factor $e^{-2z/l_s}$ accounts for the attenuation of single-scattered waves in their roundtrip to depth $z$, where $l_s$ is the scattering mean free path. $k_z^{in}(\lambda) = \sqrt{k_0(\lambda)^2 - |\mathbf{k}_{in}|^2}$ and $k_z(\lambda) = \sqrt{k_0(\lambda)^2 - |\mathbf{k}|^2}$ are the z-components of the wavevectors of the incident and reflected waves, respectively, where $k_0(\lambda) = 2\pi/\lambda$. The phase factors, $k_z^{in}(\lambda)z$ and $k_z(\lambda)z$, originate from the phase retardations due to defocusing effects in the illumination and detection pathways, respectively. The multiple-scattered wave, $\widetilde{E}_M(\mathbf{k}; \mathbf{k}_{in}, \lambda)$, includes waves scattered multiple times within the scattering medium and volumetric objects as well as those between the scattering medium and target objects.

Now that we have a set of $\widetilde{E}(\mathbf{k}; \mathbf{k}_{in}, \lambda)$ for various $\mathbf{k}_{in}$ and $\lambda$, it is possible to coherently add them in such a way that only the single-scattered waves are constructively accumulated. In fact, the summation with respect to $\lambda$ corresponds to optical coherence gating, and that with respect to $\mathbf{k}_{in}$ leads to confocal gating. Specifically, the

coherence-gated field for a target depth $z_t$ (Fig. 2d) is expressed as follows:

$$\widetilde{E}_{cg}(\mathbf{k}, z_t; \mathbf{k}_{in}) = \sum_\lambda \widetilde{E}(\mathbf{k}; \mathbf{k}_{in}, \lambda) e^{-ik_z^{in}(\lambda)z_t} e^{-ik_z(\lambda)z_t}. \quad (2)$$

Here, the phase retardations, $-k_z^{in}(\lambda)z_t$ and $-k_z(\lambda)z_t$, are introduced to remove the phase factors induced by defocused illumination to and reflection from a specific depth $z_t$ where coherence gating is to be applied. As a consequence, the images in the blue box in Fig. 2c were converted into those in the blue box in Fig. 2d.

We obtain the confocal and coherence-gated field by adding $\widetilde{E}_{cg}(\mathbf{K}, z_t; \mathbf{k}_{in})$ over $\mathbf{k}_{in}$:

$$\widetilde{E}_{ccg}(\mathbf{K}, z_t) = \sum_{|\mathbf{k}_{in}| \leq k_0\alpha} \widetilde{E}_{cg}(\mathbf{K}, z_t; \mathbf{k}_{in}) \quad (3)$$

where $\mathbf{K} = \mathbf{k} - \mathbf{k}_{in}$ is the momentum difference. The shift of the output spectrum by $-\mathbf{k}_{in}$ in Eq. (3) compensates the spectral shift of the object function in each $\widetilde{E}_{cg}(\mathbf{k}, z_t; \mathbf{k}_{in})$. It is noteworthy that the coherence-gated single-scattered wave becomes $\widetilde{E}_{cg}^S(\mathbf{k}, z_t; \mathbf{k}_{in}) \approx e^{-2z_t/l_s} \widetilde{O}(\mathbf{k} - \mathbf{k}_{in}, z_t) N_\lambda$, and confocal and coherence-gated single-scattered wave is approximately reduced to $\widetilde{E}_{ccg}^S(\mathbf{K}, z_t) \approx e^{-2z_t/l_s} \widetilde{O}(\mathbf{K}, z_t) N_\lambda N(\mathbf{K})$. Here, $N_\lambda$ is the number of wavelengths scanned by the light source, and $N(\mathbf{K})$ is the number of $\mathbf{k}_{in}$ and $\mathbf{k}$ pairs that meet the relation, $\mathbf{K} = \mathbf{k} - \mathbf{k}_{in}$. Essentially, the electric fields of the single-scattered waves $\widetilde{E}_S(\mathbf{k}; \mathbf{k}_{in}, \lambda)$ are coherently added

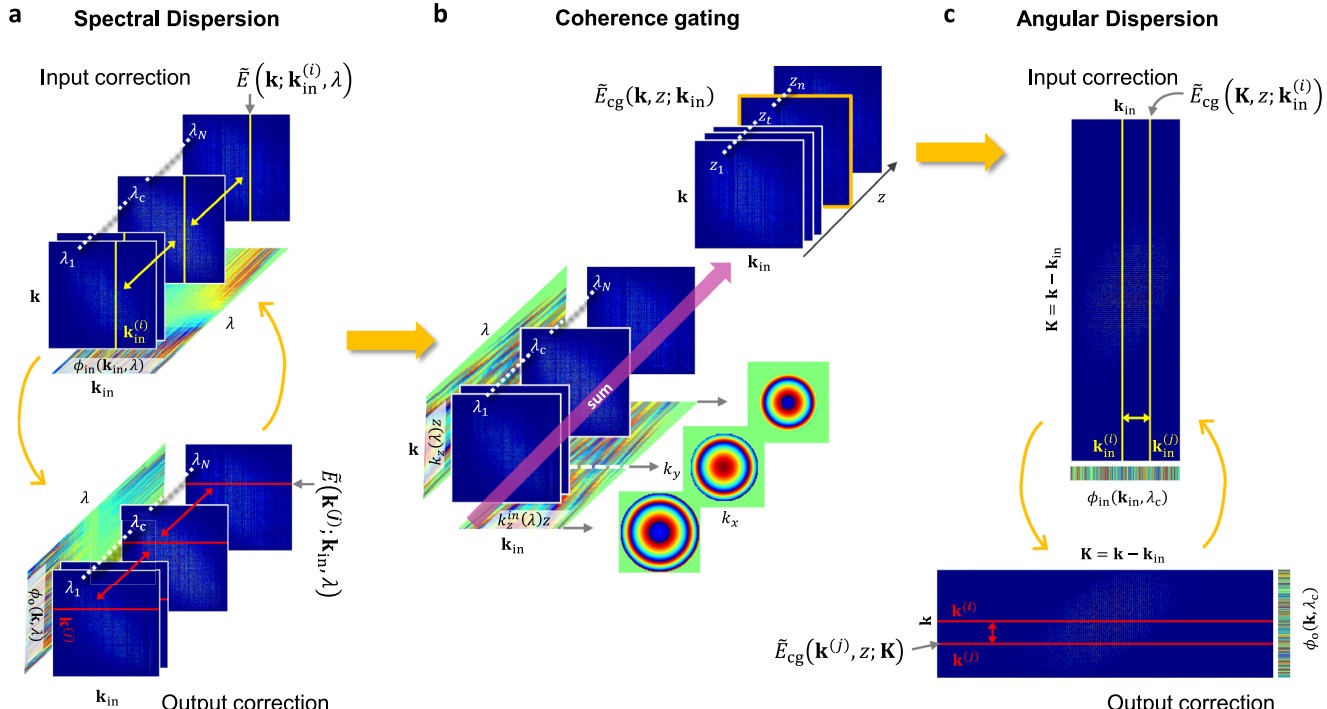

**Fig. 3 | Procedures for finding spectro-angular dispersion. a** Spectral dispersion correction. For a given $\mathbf{k}_{in}^{(i)}$, $\widetilde{E}\left(\mathbf{k};\mathbf{k}_{in}^{(i)},\lambda\right)$ of different wavelengths are marked with yellow lines. We compute correlations (yellow arrows) among the column vectors to find the input spectral dispersions. Likewise, we compute correlations (red arrows) among the row vectors indicated by the red lines, $\widetilde{E}\left(\mathbf{k}^{(j)};\mathbf{k}_{in},\lambda\right)$ of different wavelengths for a given $\mathbf{k}^{(j)}$, to find the output spectral dispersions. Input and output corrections are iterated until they are converged. **b** Coherence gating. After spectral corrections, $\widetilde{E}_{cg}(\mathbf{k},z;\mathbf{k}_{in})$ for a set of target depths $z$ are obtained by integrating $\widetilde{E}(\mathbf{k};\mathbf{k}_{in},\lambda)$ over $\lambda$ after applying a wavelength-dependent numerical propagation, $e^{-ik_z^{in}(\lambda)z}e^{-ik_z(\lambda)z}$. **c** Angular dispersion correction. For input correction,

we convert the matrix $\widetilde{E}_{cg}(\mathbf{k},z;\mathbf{k}_{in})$ to $\widetilde{E}_{cg}(\mathbf{K}=\mathbf{k}-\mathbf{k}_{in},z;\mathbf{k}_{in})$ to compensate spectrum shift of the object function. We then obtain the angle of the correlation (yellow arrow) among the column vectors indicated by the yellow lines for all the possible pairs of $\mathbf{k}_{in}^{(i)}$ and $\mathbf{k}_{in}^{(j)}$ to asymptotically find input angular dispersions. For output correction, we convert the matrix $\widetilde{E}_{cg}(\mathbf{k},z;\mathbf{k}_{in})$ to $\widetilde{E}_{cg}(\mathbf{k},z;\mathbf{K}=\mathbf{k}-\mathbf{k}_{in})$. We then obtain the angle of the correlation (red arrow) among the rows indicated by the red lines for all the possible pairs of $\mathbf{k}^{(i)}$ and $\mathbf{k}^{(j)}$ to asymptotically find output angular dispersions. Input and output corrections are iterated until they are converged.

such that $\widetilde{E}_{ccg}(\mathbf{K},z_t)$ is amplified by the factor $N_\lambda N(\mathbf{K})$. Since multiple-scattered waves are added incoherently, the SNR in the intensity image increases by the factor $\sqrt{N_\lambda N(\mathbf{K})}$, which results in the enhanced imaging depth. By taking the inverse Fourier transform of $\widetilde{E}_{ccg}(\mathbf{K},z_t)$ with respect to $\mathbf{K}$, we can obtain the coherence-gated confocal image of the object $O(\mathbf{r},z_t)$ at each depth $z_t$ (Fig. 2e).

**Correction of spectro-angular dispersion for optimal volumetric spatial resolution**
We present an algorithm for correcting the spectro-angular dispersions in the measured volumetric reflection matrix to reconstruct an object embedded within a scattering and dispersive medium with volumetric spatial resolution on par with that in the transparent medium. When probing a target object embedded within a scattering medium, the single-scattered wave experiences phase retardation $\phi_{in}(\mathbf{k}_{in},\lambda)$ on the way in and $\phi_o(\mathbf{k},\lambda)$ on its way back to the detector depending on $\lambda$, $\mathbf{k}_{in}$, and $\mathbf{k}$. In other words, an inhomogeneous scattering medium causes both spectral and angular dispersion to the waves traveling straight through the scattering medium enclosing the target objects. These resulting spectro-angular dispersions modify the single-scattered wave in the measured electric field $\widetilde{E}(\mathbf{k};\mathbf{k}_{in},\lambda)$:

$$\widetilde{E}_S(\mathbf{k};\mathbf{k}_{in},\lambda) = e^{i\phi_{in}(\mathbf{k}_{in},\lambda)}e^{i\phi_o(\mathbf{k},\lambda)}\int e^{-2z/l_s}\widetilde{O}(\mathbf{k}-\mathbf{k}_{in},z)e^{ik_z^{in}(\lambda)z}e^{ik_z(\lambda)z}dz.$$

(4)

Since the multiple-scattered wave is transformed into another random speckle field by the dispersion, there is little change in their

contribution to confocal and coherence gating. $\phi_{in}(\mathbf{k}_{in},\lambda)$ and $\phi_o(\mathbf{k},\lambda)$ undermine the optimal accumulation of single-scattered waves in forming a coherence-gated confocal image, leading to a reduction in imaging depth. Equation (4) assumes that $\phi_{in}(\mathbf{k}_{in},\lambda)$ and $\phi_o(\mathbf{k},\lambda)$ are the same within the volume of interest.

Here, we introduce a method to find both spectral and angular dispersion in situ and reconstruct the object function $O(x,y,z)$ of a volumetric object (see Supplementary Information Section III for the complete derivation). The dispersion correction method comprises two major steps. The first step is to find the spectral dispersion with respect to $\lambda$ in $\phi_{in}(\mathbf{k}_{in},\lambda)$ for each fixed $\mathbf{k}_{in}$ (Fig. 3a). Specifically, one can asymptotically find $\phi_{in}(\mathbf{k}_{in},\lambda) - \phi_{in}(\mathbf{k}_{in},\lambda_c)$, which is the relative dispersion with respect to $\lambda_c$, the center wavelength, by the angle of the correlation, $\langle\widetilde{E}(\mathbf{k};\mathbf{k}_{in},\lambda)\widetilde{E}^*(\mathbf{k};\mathbf{k}_{in},\lambda_c)\rangle_{\mathbf{k}}$. $\langle\rangle_{\mathbf{k}}$ represents the summation of the elements within the bracket with respect to $\mathbf{k}$. Yellow lines in Fig. 3a indicate $\widetilde{E}(\mathbf{k};\mathbf{k}_{in},\lambda)$ for various $\lambda$ at a fixed $\mathbf{k}_{in}$, and yellow arrows depicts the correlation among them. Likewise, the relative dispersion, $\phi_o(\mathbf{k},\lambda) - \phi_o(\mathbf{k},\lambda_c)$, can be found by the correlation, $\langle\widetilde{E}(\mathbf{k};\mathbf{k}_{in},\lambda)\widetilde{E}^*(\mathbf{k};\mathbf{k}_{in},\lambda_c)\rangle_{\mathbf{k}_{in}}$, for each $\mathbf{k}$. Red lines in Fig. 3a indicate $\widetilde{E}(\mathbf{k};\mathbf{k}_{in},\lambda)$ for various $\lambda$ at a fixed $\mathbf{k}$, and red arrows depicts the correlation among them. This correlation analysis is performed for all wavelengths with respect to a center wavelength $\lambda_c$, and the identified dispersions are corrected for each $\widetilde{E}(\mathbf{k};\mathbf{k}_{in},\lambda)$.

Once the spectral dispersion has been corrected, the spectro-angular dispersion problem has been reduced to finding the angular dispersion, i.e. finding $\phi_{in}(\mathbf{k}_{in},\lambda_c)$ and $\phi_o(\mathbf{k},\lambda_c)$ with respect to $\mathbf{k}_{in}$ and $\mathbf{k}$, respectively. To perform this second step, we construct a coherence-gated electric field $\widetilde{E}_{cg}(\mathbf{k},z;\mathbf{k}_{in})$ by coherently adding the

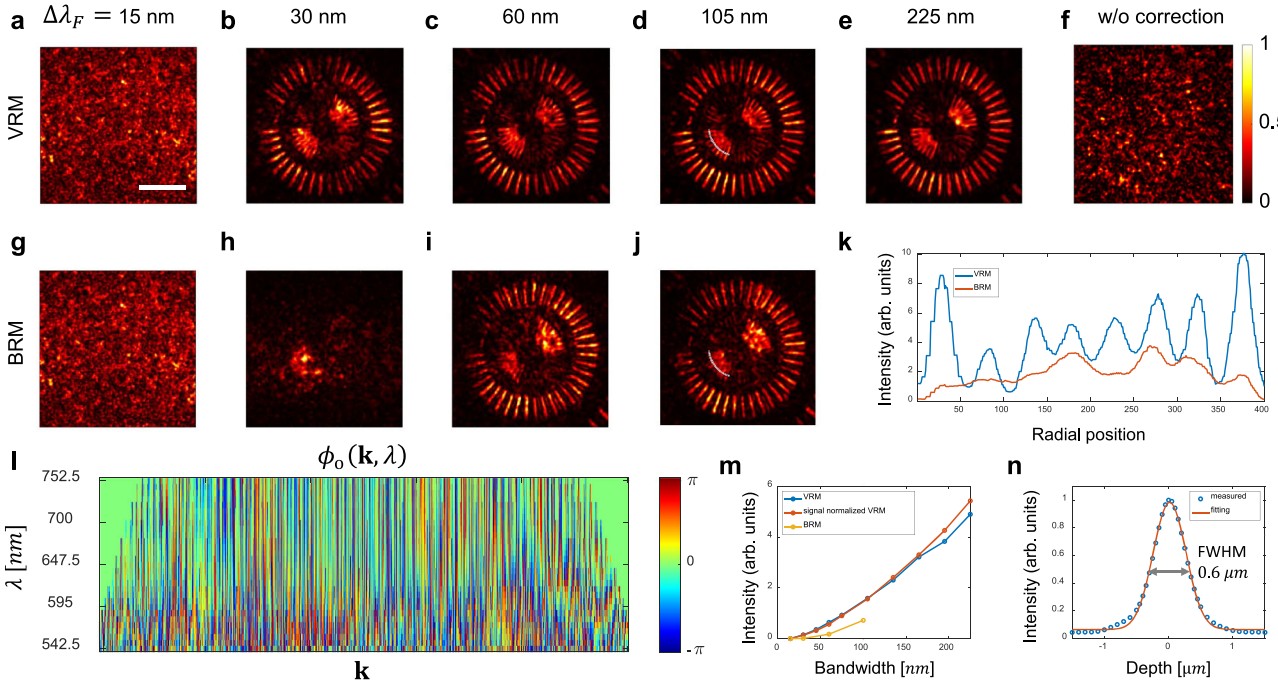

**Fig. 4 | Correction of spectro-angular dispersion for optimal volumetric resolution. a–e** Reconstructed VRM (Volumetric Reflection Matrix) images for spectral bandwidth coverage $\Delta\lambda_F$ of 15, 30, 60, 105, and 225 nm after spectro-angular dispersion correction, respectively. Scale bar, 5 $\mu m$. **f** Reconstructed VRM images for $\Delta\lambda_F = 225 nm$ without dispersion correction. **g–j** Reconstructed images in experiments with broadband source whose $\Delta\lambda_F$ is the same as in **a–d**, respectively. Therefore, spectral dispersion could not be corrected. Color bar, normalized intensity by the maximum intensity in each image (intensity images). **k** Radial line plots of **d** (blue) and **j** (red) along the white line. **l** Output spectro-angular dispersion $\phi_o(\mathbf{k},\lambda)$ with respect to $\lambda$ and $\mathbf{k}$ in the case of $\Delta\lambda_F = 225 nm$. Color bar, phase in radians. **m** Signal intensity of VRM (blue), signal normalized VRM (red) and BRM (Broadband Reflection Matrix, yellow) with respect to $\Delta\lambda_F$. **n** Intensity(circles) versus imaging depth. Line shows the Gaussian fit of data. Full width at half maximum is 0.6 $\mu m$, which corresponds to depth resolution.

spectral dispersion corrected $\widetilde{E}(\mathbf{k};\mathbf{k}_{in},\lambda)$ with respect to $\lambda$ according to Eq. (2) (Fig. 3b). A wavelength-dependent numerical propagation, $e^{-ik_z^{in}(\lambda)z}e^{-ik_z(\lambda)z}$ is applied to $\widetilde{E}(\mathbf{k};\mathbf{k}_{in},\lambda)$ prior to the spectral integration. For compensating spectrum shift, we convert the matrix $\widetilde{E}_{cg}(\mathbf{k},z;\mathbf{k}_{in})$ to $\widetilde{E}_{cg}(\mathbf{K}=\mathbf{k}-\mathbf{k}_{in},z;\mathbf{k}_{in})$. We then obtain the angle of the correlation $\langle\widetilde{E}_{cg}(\mathbf{K},z;\mathbf{k}_{in}^{(1)})\widetilde{E}_{cg}^*(\mathbf{K},z;\mathbf{k}_{in}^{(2)})\rangle_{\mathbf{K}}$ for all the possible pairs of $\mathbf{k}_{in}^{(1)}$ and $\mathbf{k}_{in}^{(2)}$ to asymptotically find $\phi_{in}(\mathbf{k}_{in}^{(2)},\lambda_c)-\phi_{in}(\mathbf{k}_{in}^{(1)},\lambda_c)$. Yellow lines in Fig. 3c indicate $\widetilde{E}_{cg}(\mathbf{K},z;\mathbf{k}_{in}^{(1)})$ and $\widetilde{E}_{cg}(\mathbf{K},z;\mathbf{k}_{in}^{(2)})$ for the representative $\mathbf{k}_{in}^{(1)}$ and $\mathbf{k}_{in}^{(2)}$, and the yellow arrow depicts their correlation. Likewise, we convert the matrix $\widetilde{E}_{cg}(\mathbf{k},z;\mathbf{k}_{in})$ to $\widetilde{E}_{cg}(\mathbf{k},z;\mathbf{K}=\mathbf{k}-\mathbf{k}_{in})$ for output correction. Then, we compute the correlation $\langle\widetilde{E}_{cg}(\mathbf{k}^{(1)},z;\mathbf{K})\widetilde{E}_{cg}^*(\mathbf{k}^{(2)},z;\mathbf{K})\rangle_{\mathbf{K}}$ for all the possible pairs of $\mathbf{k}^{(1)}$ and $\mathbf{k}^{(2)}$ to find $\phi_o(\mathbf{k}^{(2)},\lambda_c)-\phi_o(\mathbf{k}^{(1)},\lambda_c)$. Red lines in Fig. 3c indicate $\widetilde{E}_{cg}(\mathbf{k}^{(1)},z;\mathbf{K})$ and $\widetilde{E}_{cg}(\mathbf{k}^{(2)},z;\mathbf{K})$ for the representative $\mathbf{k}^{(1)}$ and $\mathbf{k}^{(2)}$, and red arrow depicts their correlation. After these two steps, we obtain $\phi_{in}(\mathbf{k}_{in},\lambda)$ and $\phi_o(\mathbf{k},\lambda)$ for each $\lambda$, $\mathbf{k}_{in}$, and $\mathbf{k}$. We perform iterations within each step and between the two steps to increase the precision of the correction, which is required due to the presence of strong multiple scattering and two-way angular dispersions. After completing the spectro-angular dispersion correction, we obtain confocal and coherence-gated electric field $\widetilde{E}_{ccg}(\mathbf{K},z_t)$ based on Eq. (3) and take its inverse Fourier transform to obtain the object function (see Supplementary Information Section III for the object function reconstruction after spectro-angular dispersion correction).

We experimentally demonstrated the spectro-angular dispersion correction and proved its benefit over conventional volumetric imaging in terms of imaging depth and spatial resolution. A resolution target (see Supplementary Information Section V) was placed under a scattering medium made of randomly dispersed 1-μm-diameter polystyrene beads in PDMS. The scattering mean free path

$(l_s)$ of the scattering medium was 56.1 μm and 83.1 μm for 535 nm and 760 nm, respectively. The anisotropic factor g was 0.91, resulting in the transport mean free path, $l_t = 11 l_s$. In addition, the scattering medium was covered with a 50-μm-thick, rough-surfaced plastic layer with strong spectro-angular dispersion. Ballistic wave attenuation by multiple scattering and PSF attenuation by the spectro-angular dispersion are often quantified by the scattering mean free path and the Strehl ratio, respectively. Since they jointly distort the PSF, we estimated the effective optical thickness ($l_{eff}$) of the scattering medium based on the total attenuation of peak PSF intensity, which amounts to 9.4 times the scattering mean free path (see Supplementary Information Section IV for details). Subsequently, we took a set of electric field images $E(\mathbf{r};\boldsymbol{\theta}_{in},\lambda)$ to construct the volumetric reflection matrix. We applied the spectro-angular dispersion correction method described above while increasing the full spectral bandwidth coverage $\Delta\lambda_F$ from 15 to 225 nm with respect to the center wavelength $\lambda_c = 647.5$ nm. For example, $\Delta\lambda_F = 105 nm$ means that we chose $E(\mathbf{r};\boldsymbol{\theta}_{in},\lambda)$ for the incident wavelength $\lambda$ to be within $647.5 \pm 52.5$ nm in the VRM image construction. Fig. 4a–e show the reconstructed VRM images for 15, 30, 60, 105, and 225 nm $\Delta\lambda_F$, respectively. Fig. 4l shows the spectro-angular dispersion $\phi_o(\mathbf{k},\lambda)$ with respect to $\lambda$ and $\mathbf{k}$ for $\Delta\lambda_F = 225$ nm. Fig. 4f shows the reconstructed image without correction of the spectro-angular dispersion. The target image cannot be reconstructed without dispersion correction owing to severe spectro-angular dispersion. Notably, the target image could not be reconstructed when $\Delta\lambda_F = 15$ nm due to insufficient coherence gating for rejection of multiple scattering backgrounds. As the bandwidth increases, the target image becomes clearer, supporting that the single-scattered waves have effectively been accumulated in the broad wavelength range.

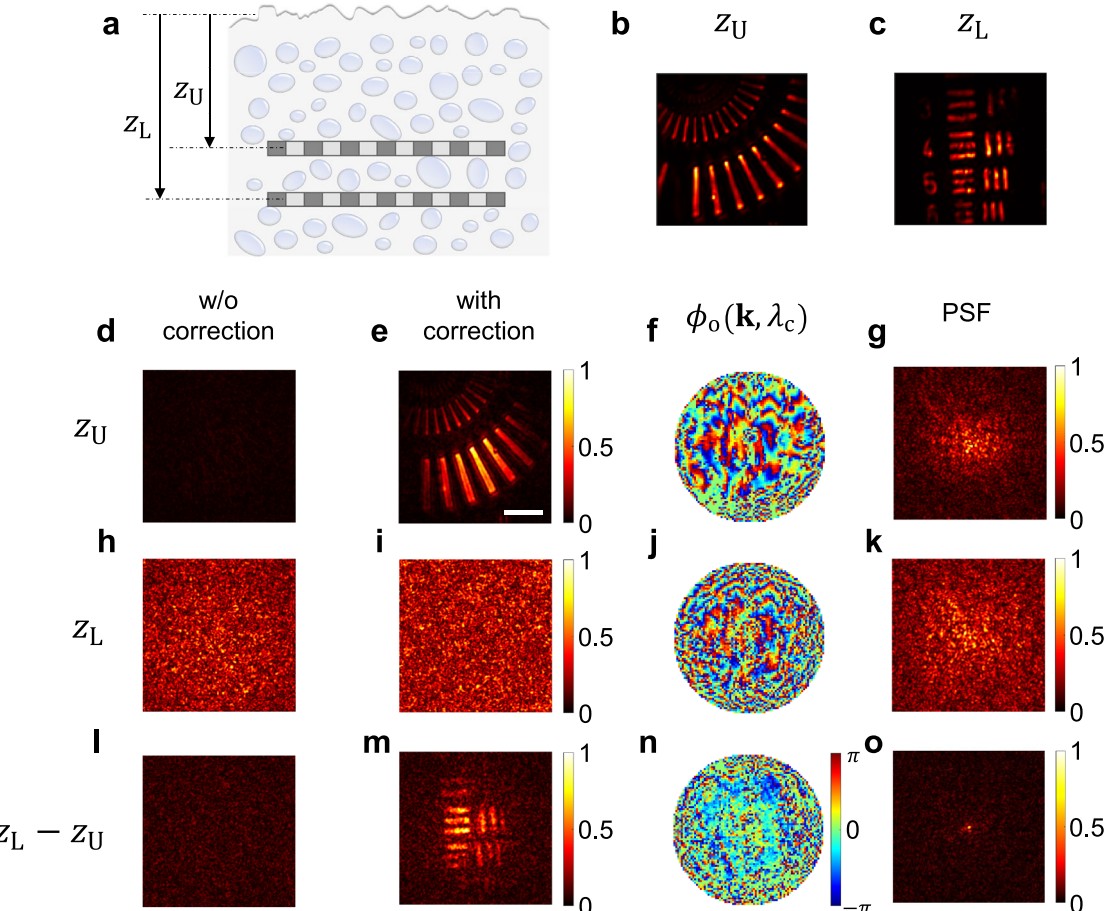

**Fig. 5 | Progressive depth imaging. a** Layout of bilayer sample. Siemens star-like target (**b**) and USAF target (**c**) embedded in scattering medium 80 μm apart in depth. The aberration medium is placed above the sample. **b**, **c** Ground-truth images for each target. **d**, **e** Coherence-gated confocal images of the upper layer, $E_{ccg}(\mathbf{r}, z_t = z_U)$, before and after dispersion correction, respectively. Color bar, normalized intensity by the maximum intensity at **e**. Scale bar, 10 μm. **f**, **g** Output angular dispersion of upper layer, $\phi_o^U(\mathbf{k}, \lambda_c)$ at center wavelength and its point-spread-function, respectively. **h**, **i** Same as **d** and **e**, but of lower layer. Color bar, normalized **i**ntensity by the maximum intensity at **i**. **j**, **k** Same as **f**, **g**, but of lower layer. **l**, **m** Same as **h**, **i**, but after application of spectro-angular dispersion of upper layer. Color bar, normalized intensity by the maximum intensity at **m**. **n**, **o** Same as **f** and **g**, but after application of spectro-angular dispersion of upper layer. Color bars in **g**, **k**, and **o**: normalized intensity by the maximum intensity in each image. Color bar in **n**: phase in radians for **f**, **j**, and **n**.

To further prove this claim, we calculated the signal intensity depending on $\Delta\lambda_F$. The signal intensity was obtained by subtracting the average intensity of the region without the target from that of the region with the target. Theoretically, the single scattering intensity increases quadratically with $N_\lambda$ due to coherent addition, but the increase in the obtained signal intensity slightly deviates from the quadratic increase (blue dots in Fig. 4m). This is because the single scattering intensity at shorter wavelengths is smaller due to the shorter scattering mean free path (see Supplementary Information Section VI). We normalized $\tilde{E}(\mathbf{k}; \mathbf{k}_{in}, \lambda)$ such that the single-scattered wave in each $\lambda$ had the same average intensity. This makes the intensity increase quadratically with $\Delta\lambda_F$ (red dots in Fig. 4m), which agrees well with the theoretical prediction. The quadratic increase in the signal intensity with $\Delta\lambda_F$ supports that the single-scattered waves were optimally accumulated by correcting the spectro-angular dispersion.

We compared the VRM method with the previously developed broadband reflection matrix (BRM) method in which a broadband light source is used to scan illumination angles[11]. Therefore, the spectral dispersion cannot be corrected in BRM. Only angular dispersions can be corrected with respect to the center wavelength of the broadband source. Since our supercontinuum source provides a maximum bandwidth of 100 nm, we could experimentally determine the BRM for up to $\Delta\lambda_F = 100$ nm bandwidth. Fig. 4g–j show the reconstructed

images from BRM for the same $\Delta\lambda_F$ as in Fig. 4a–d, respectively. For $\Delta\lambda_F = 30$ nm bandwidth, the broadband measurement could not reconstruct the target image (Fig. 4h), whereas the target image could be reconstructed with the VRM method (Fig. 4b). Even when the target image could be reconstructed with the broadband method ($\Delta\lambda_F \geq 60$ nm), the image quality of the BRM method was much worse than that of the VRM method. This is evident in Fig. 4k, which presents the radial line plots along the white line in Fig. 4d and j. The finest features of the target can be observed only with the VRM method (blue curve), but not with the BRM (red curve). As the bandwidth increases, the signal intensity of the BRM method (yellow dots in Fig. 4m) increases little compared to that of the VRM method (blue dots). All these results support the importance of correcting spectral dispersion as well as angular dispersion, especially in the case of ultra-broad spectral bandwidth.

The accurate correction of spectro-angular dispersion ensures optimal axial and lateral resolution determined by the full bandwidth $\Delta\lambda_F$ and numerical aperture of the system. For measuring the axial resolution of our system, we performed numerical propagation to different target depths and obtained the intensity of the VRM image at each target depth. The measured intensity versus the target depth for $\Delta\lambda_F = 225$ nm (circles), and the Gaussian fit of the data are shown in Fig. 4n. The measured depth resolution was 0.6 μm, and it shows a

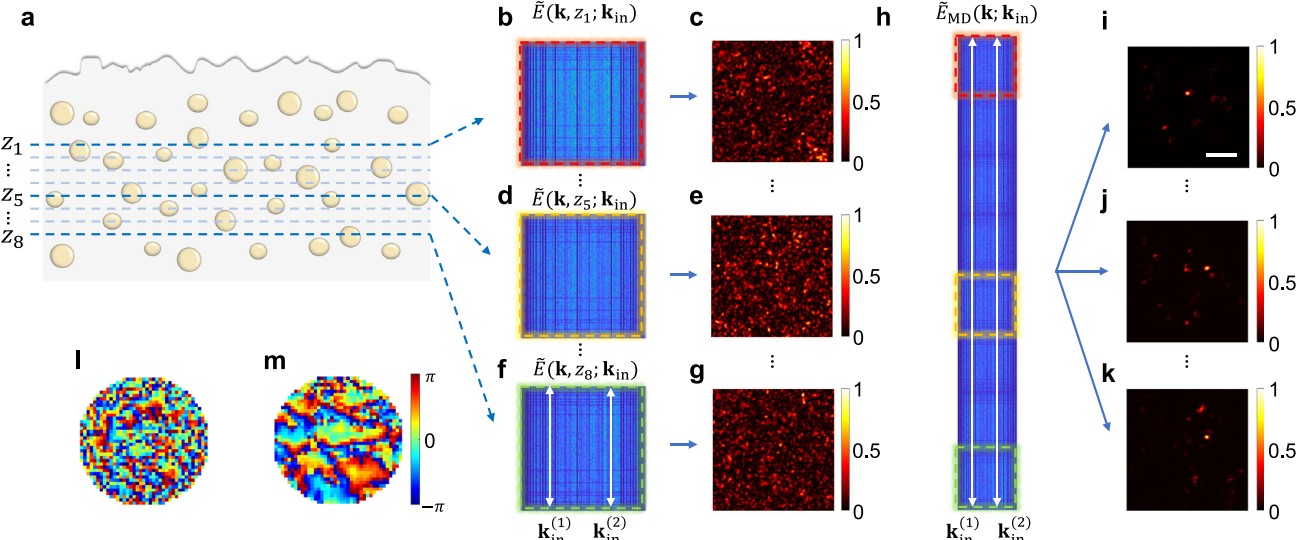

**Fig. 6 | Forming a multi-depth reflection matrix and volumetric dispersion correction. a** Layout of volumetric sample. $TiO_2$ particles are randomly dispersed in PDMS. Aberration medium is placed above sample. **b**, **d**, and **f** Coherence-gated reflection matrices $\tilde{E}_{cg}(\mathbf{k}, z; \mathbf{k}_{in})$ of three representative depths. **c**, **e**, and **g** Object images obtained from respective reflection matrices in **b**, **d**, and **f** after dispersion correction for each depth. **h** Extended-depth coherence-gated reflection matrix,

$\tilde{E}_{MD}(\mathbf{k}; \mathbf{k}_{in})$, constructed by appending $\tilde{E}_{cg}(\mathbf{k}, z; \mathbf{k}_{in})$ of eight adjacent depths. **i**–**k** Object images of each depth after dispersion correction with $\tilde{E}_{MD}(\mathbf{k}; \mathbf{k}_{in})$. **l**, **m** Input and output spectro-angular dispersions obtained with $\tilde{E}_{MD}(\mathbf{k}; \mathbf{k}_{in})$. Scale bar, 10 µm. Color bar, normalized intensity by the maximum intensity in each image (intensity images). Color bar in **m**: phase in radians for **l**, **m**.

good agreement with the theoretical prediction estimated in the transparent medium (0.59 $\mu m$). The upper bound of the measured lateral resolution was 0.41 $\mu m$, close to the diffraction-limited resolution of 0.38 $\mu m$.

**Progressive depth imaging**

By exploiting the imaging method yielding coherent volumetric information, we could achieve an unprecedented imaging depth at which multiple scattering and dispersion are so strong that a target cannot be reconstructed with conventional deep-tissue imaging methods. Specifically, we propose two unique approaches taking the advantage of the volumetric reflection matrix—progressive depth imaging and volumetric dispersion correction.

The progressive depth imaging approach uses the dispersion information of a relatively shallow depth as a priori knowledge to deal with scattering and angular dispersion at deeper depths at which direct access for reconstructing object images is impossible. Specifically, we apply the correction of the spectro-angular dispersions obtained at a shallower depth to the measured volumetric matrix. This effectively reduces the complexity of the scattering medium in the reconstruction of the image at a deeper depth. To experimentally demonstrate the proposed concept, we prepared a sample with two target layers embedded inside a scattering medium composed of randomly dispersed 1 µm diameter polystyrene beads in PDMS. The scattering mean free path ($l_s$) of the scattering medium was 48.5 µm. The scattering medium was covered with a 50 µm thick, rough-surfaced plastic layer with strong spectro-angular dispersion (Fig. 5a) (see Supplementary Information Section IV for the optical thickness of the scattering medium). Figure 5b, c show the ground-truth target images at $z_U$ and $z_L$ depths, respectively, taken in the absence of the scattering and aberrating medium. The two target layers were $z_L - z_U = 80$ $\mu m$ apart. The effective optical thickness ($l_{eff}$) of the complex medium was ~ 6 $l_s$ at $z_U$ and ~ 9 $l_s$ at $z_L$.

The volumetric reflection matrix $\tilde{E}(\mathbf{k}; \mathbf{k}_{in}, \lambda)$ was obtained for the prepared sample over the volume $40 \times 40 \times 100$ $\mu m^3$. In this implementation, we adopted a swept-source laser (Superlum BS-

840-1-HP, spectral range: 803–878 nm) that emits an approximately monochromatic wave at each wavelength to ensure a volumetric reflection matrix with a wide depth range (see Methods for experimental setup details). The estimated depth resolution was ~3.0 µm at the sample based on the spectral range of the laser. First, we tried to reconstruct the object at $z_t = z_U$. Figure 5d, e show the coherence-gated confocal images $E_{ccg}(\mathbf{r}, z_t = z_U)$ before and after dispersion correction, respectively. The object image was successfully reconstructed with the correction method. Figure 5f shows the output angular dispersion of $z_t = z_U$, $\phi_o^U(\mathbf{k}, \lambda_c)$, and its point-spread-function (PSF) (Fig. 5g) represents the complexity of the angular dispersion. In the next step, we reconstructed the image of a target at $z_t = z_L$. Due to additional scattering and dispersion at the increased depth, the dispersion correction method failed to work. Consequently, the object information could not be recovered even after the dispersion correction (Fig. 5i), let alone before the correction (Fig. 5h).

To retrieve the object image at $z_t = z_L$, we applied the spectro-angular dispersion, $\phi_{in}^U(\mathbf{k}_{in}, \lambda)$ and $\phi_o^U(\mathbf{k}, \lambda)$ of the upper layer to the original volumetric reflection matrix, i.e. $\tilde{E}_c(\mathbf{k}; \mathbf{k}_{in}, \lambda) = e^{-i\phi_o^U(\mathbf{k}, \lambda)} \tilde{E}(\mathbf{k}; \mathbf{k}_{in}, \lambda) e^{-i\phi_{in}^U(\mathbf{k}_{in}, \lambda)}$. By performing dispersion correction for this corrected matrix in the process of obtaining $E_{ccg}(\mathbf{r}, z_t = z_L)$, we successfully recovered the object image at $z_t = z_L$ (Fig. 5m). Fig. 5n shows the identified dispersion map $\phi_o^{L-U}(\mathbf{k}, \lambda_c)$, which corresponds to the dispersion difference between $z_U$ and $z_L$. Therefore, its PSF distortion (Fig. 5o) was much weaker than that of the upper layer (Fig. 5g). The net dispersion up to depth $z_L$ can be obtained by adding the dispersion of the upper layer and that of the difference, i.e. $\phi_o^L(\mathbf{k}, \lambda_c) = \phi_o^U(\mathbf{k}, \lambda_c) + \phi_o^{L-U}(\mathbf{k}, \lambda_c)$. As shown in Fig. 5j, k, the dispersion map and PSF associated with $z_L$ show much more severe distortion than those of the upper layer. This explains why dispersion correction did not work when $z_L$ was directly accessed. In fact, Fig. 5n, o clearly show that correcting the upper layer dispersion effectively converted the scattering medium less dispersive for imaging an object underneath.

We present the theoretical basis to quantitatively assess the benefit of progressive depth imaging. The key to finding the input

dispersion is to obtain $\phi_{in}(\mathbf{k}_{in}^{(2)}, \lambda_c) - \phi_{in}(\mathbf{k}_{in}^{(1)}, \lambda_c)$ from the angle of the correlation $\langle \widetilde{E}_{cg}(\mathbf{K} + \mathbf{k}_{in}^{(1)}, z_t; \mathbf{k}_{in}^{(1)}) \widetilde{E}_{cg}^*(\mathbf{K} + \mathbf{k}_{in}^{(2)}, z_t; \mathbf{k}_{in}^{(2)}) \rangle_{\mathbf{K}}$. The precision level of this process is strongly affected by the presence of multiple scattering and output dispersion. In fact, dispersion correction can work when $C_{rel} \approx \frac{S}{M} \zeta \sqrt{N}$ (which defines the relative contribution of single- to multiple-scattered waves to this correlation) is greater than a certain threshold (see Supplementary section VII for full derivation and its experimental validation). Here, $S$ and $M$ are the average intensities of single- and multiple-scattered waves at each detection channel, respectively, and $N$ is interpreted as the total number of elements involved in the cross-correlation of pupil functions. $\zeta$ represents the complexity of both input and output angular dispersions, and $\zeta = 1$ when both input and output dispersions do not exist.

Direct imaging at depth $z_L$ was impossible because $|\zeta_L|$ was too small due to the complexity of $\phi_o^L$ and $\phi_{in}^L$ such that $C_{rel}$ was below a certain threshold. The progressive depth imaging approach increases $|\zeta_L|$ to $|\zeta_{L-U}|$, which is determined by $\phi_o^L(\mathbf{k}, \lambda_c) - \phi_o^U(\mathbf{k}, \lambda_c)$ and $\phi_{in}^L(\mathbf{k}, \lambda_c) - \phi_{in}^U(\mathbf{k}, \lambda_c)$, by compensating for upper layer dispersion. In our demonstration in Fig. 5, $|\zeta_L| = 0.0012$ and $|\zeta_{L-U}| = 0.0388$ such that $C_{rel}$ increased ~32 times. In principle, $|\zeta|$ can be enhanced close to 1 if dispersion is found while increasing the depth gradually, suggesting that $C_{rel}$ can be enhanced by a factor of $1/|\zeta|$. In summary, progressive depth imaging enables us to make good use of a single scattering correlation as if the single scattering signal was initially higher by removing upper layer dispersion. This translates into an effective increase in the single scattering intensity from $|\zeta| S$ to $S$ by a factor of $1/|\zeta|$.

## Volumetric dispersion correction

As discussed above, finding spectro-angular dispersion depends on the correlation of single-scattered waves, which is hampered by the presence of multiple-scattered waves. Imaging fidelity is governed by the parameter $C_{rel} = \frac{S}{M} \zeta \sqrt{N}$, which means that the achievable imaging depth highly depends on the number of output channels in $N$, i.e. the number of detection pixels. Here, we propose the use of all the single-scattered waves from a target object volume. In contrast to previously presented approaches where single-scattered waves from a particular depth section are used for dispersion correction, this proposed approach exploits single-scattered waves detected over a certain volume. Therefore, the number of detection pixels is greatly increased from a 2D section to a 3D volume, thereby enhancing the fidelity of image reconstruction.

To verify the proposed concept, we prepared a scattering medium composed of randomly dispersed $TiO_2$ particles with 1 μm mean diameter in PDMS (Fig. 6a). The scattering mean free path ($l_s$) of the scattering medium was 62.4 μm. The anisotropic factor g was 0.38, resulting in the transport mean free path, $l_t = 1.6 l_s$. The scattering medium was covered with a rough-surfaced plastic layer with complex dispersion, and the maximum effective optical thickness ($l_{eff}$) was 6.9 $l_s$ (see Supplementary Information Section IV for the optical thickness of the scattering medium). We measured a volumetric reflection matrix $\widetilde{E}(\mathbf{k}; \mathbf{k}_{in}, \lambda)$ and obtained coherence-gated reflection matrices $\widetilde{E}_{cg}(\mathbf{k}, z_t; \mathbf{k}_{in})$ for eight different depths (Fig. 6b, d, and f show three representative depths). For each single-depth matrix, the multiple scattering and dispersion were so severe that $C_{rel}$ was below the threshold for successful dispersion compensation. Consequently, the object images could not be recovered at any of the depths (Fig. 6c, e, and g). To increase the fidelity of single scattering correlation, we merged multiple coherence-gated fields from different depths to form a multi-depth coherence-gated electric field:

$$\widetilde{E}_{MD}(\mathbf{k}; \mathbf{k}_{in}) = \left[ \widetilde{E}_{cg}(\mathbf{k}, z_1; \mathbf{k}_{in}), \widetilde{E}_{cg}(\mathbf{k}, z_2; \mathbf{k}_{in}), \cdots, \widetilde{E}_{cg}(\mathbf{k}, z_8; \mathbf{k}_{in}) \right] \quad (5)$$

Coherence-gated electric fields from multiple depths $z_1, z_2, \cdots, z_8$ were appended from one another to form an extended-depth coherence-gated field. Fig. 6h shows $\widetilde{E}_{MD}(\mathbf{k}; \mathbf{k}_{in})$ constructed by appending $\widetilde{E}_{cg}$ from eight different depths. Subsequently, we computed the correlation, $\langle \widetilde{E}_{MD}(\mathbf{K}; \mathbf{k}_{in}^{(1)}) \widetilde{E}_{MD}^*(\mathbf{K}; \mathbf{k}_{in}^{(2)}) \rangle_{\mathbf{K}}$, to obtain $\phi_{in}(\mathbf{k}_{in}^{(2)}, \lambda_c) - \phi_{in}(\mathbf{k}_{in}^{(1)}, \lambda_c)$. We performed a similar multi-depth process to estimate $\phi_o(\mathbf{k}^{(2)}, \lambda_c) - \phi_o(\mathbf{k}^{(1)}, \lambda_c)$. This enabled us to obtain $\phi_{in}(\mathbf{k}_{in}, \lambda_c)$ and $\phi_o(\mathbf{k}, \lambda_c)$ (Fig. 6l and m, respectively) and retrieve object images (Fig. 6i–k) at depths at which the single-depth approach fails to work (Supplementary Information Section VIII supports that the images in Fig. 6i–k are the typical reflectance images of TiO2 particles). It should be noted that this process can be applied over the depth range where the spectro-angular dispersion is similar. This requirement resembles the 2D dispersion correction within the iso-planatic patch in the lateral plane.

When multiple-scattered waves of different depths are independent of one another, the multiple scattering correlation is proportional to $\sqrt{N_z}$, where $N_z$ is the number of independent merged depths in our volumetric dispersion correction approach. By contrast, the single scattering correlation is proportional to $N_z$. Therefore, the fidelity of the single scattering correlation is effectively increased from $C_{rel} = \frac{S}{M} \zeta \sqrt{N}$ to $C_{rel}^{MD} = C_{rel} \sqrt{N_z}$. In our example in Fig. 6, we combined eight different depths with 1.7 μm intervals over the depth range of 12 μm where the spectro-angular dispersion was approximately the same. To determine the effective increase of the fidelity, we computed the amplitude of coherently added multiple-scattered waves of different depths. Due to the non-zero correlation among multiple-scattered waves of different depths, the multiple scattering amplitude was increased by the factor $\sqrt{11.4}$, which is slightly greater than $\sqrt{8}$, the value expected when the multiple-scattered waves are completely uncorrelated. Therefore, there was an effective increase of the detection channels by a factor of 5.6 in the exploitation of the multi-depth matrix merging 8 different depths.

## Discussion

We presented a general volumetric image reconstruction framework embracing both the wavelengths and the angles of propagating waves. This method provides the most fundamental level of understanding the image formation compared to conventional imaging modalities considering the integral coverage of either wavelengths or angles, or both. In this respect, this study marks an important milestone in the field of optical imaging. On this basis, we developed an algorithm removing both the spectral and angular dispersion set by wavelength- and angle-dependent phase retardations, respectively, induced by a complex scattering medium. We constructed an experimental system that scans the wavelength range from 535 to 760 nm (spectral band-width: 225 nm) and illumination angles of up to 1.0 NA. By applying the developed algorithm, we achieved diffraction-limited ultrahigh volumetric resolution (lateral: 0.41 μm, axial: 0.60 μm) deep within a scattering medium where previous approaches only dealing with either spectral or angular dispersion lose resolving power.

Furthermore, we proposed two deep imaging approaches exploiting coherent volumetric information that take the imaging depth beyond those of previous 2D-image-based methods. The first was the progressive depth imaging approach which uses the spectro-angular dispersion found at a shallower depth to convert the scattering medium less dispersive for imaging at deeper depths. This enabled us to reduce the degree of dispersion, equivalent to a 32-fold increase in the effective single scattering intensity. Essentially, this progressive approach eliminates spectro-angular dispersion such that the achievable imaging depth is solely determined by multiple scattering backgrounds. The second approach was to construct a multi-depth

volumetric reflection matrix and coherently use the signal waves from a volume in coping with multiple scattering. In our demonstration, we could use 5.6 times more signal than 2D approaches, leading to increased imaging depth.

The working condition of the proposed approaches is set by the parameter $C_{rel} = \frac{S}{M}\zeta\sqrt{N}$. For our method to work, the single-scattering intensity $S$ set by the reflectance and the depth of the target object, multiple scattering intensity $M$, and the complexity of the spectro-angular dispersion $\zeta$ should meet the criteria such that $C_{rel}$ should be larger than a certain threshold. The volumetric reflection matrix approach becomes more effective with increasing 2D depth sections in the covered volume. However, this involves taking more images. Since both the wavelength and illumination angle should be scanned for each complex widefield image recording, the required number of images is $N_\lambda N_\theta$, where $N_\lambda$ and $N_\theta$ are the numbers of scanned wavelengths and angles, respectively. In the presented experiment, the camera's frame rate (300 Hz) is a major limiting factor in the acquisition time. In the case of the experiments using the swept-source laser, the $40 \times 40 \times 89 \mu m^3$ volume at the sample (0.4 NA) can be obtained with $N_\lambda = 30$ and $N_\theta = 1245$, which takes ~2 min for obtaining the VRM. The computation time for finding the spectro-angular dispersion was ~10 s when a desktop computer equipped with an intel i9-11900k processor was used. If a high-speed camera with 6000 frames per second is used, the acquisition time can be reduced to 6 s. To make the method compatible with in vivo imaging, the acquisition time can be shortened by downsampling $N_\lambda$ and $N_\theta$ depending on the degrees of spectro-angular dispersion and multiple scattering.

In the context of solving the inverse scattering problem, previous studies[25–27] including the multispectral transmission matrix recording[26] showed the feasibility of inverting multiple scattering from one side of the scattering layer to the other side. However, the transmission matrix can be measured only when a detector is placed on the opposite side of the scattering layer where a target object is placed. Its measurements for the scattering medium covering an embedded target remains to be addressed. In our study, we measured the volumetric reflection matrix and resolved the two-way (roundtrip) distortion of the ballistic components using an ingenious algorithm as the simple matrix inverse cannot be applied to finding embedded targets. To a certain extent, our approach found the transmission matrix in situ for imaging embedded targets up to its ballistic components. In fact, it can be extended to measuring the position-dependent spectro-angular dispersions[11,12], which is equivalent to finding multiple scattering components of the transmission matrix. In this respect, our study marks an important step from the previous transmission matrix study toward solving the inverse problem of an embedded target exploiting both ballistic and multiple scattering. The coherent use of volumetric imaging proposed in our study will open new opportunities for deep imaging. The volumetric reflection matrix provides the most extensive collection of light-sample interaction information conceivable. This can potentially offer a powerful database to solve a high-order inverse problem aiming at the deterministic use of multiple scattering in situ[3,28,29]. With the increase in computation power, the ability to resolve a complex inverse problem is growing fast, and the existence of vast training data will enhance its feasibility. The use of spatio-spectral information can also offer new avenues for studying meso-scopic wave correlations and their use for wave focusing and light energy delivery[25–27,30–32].

## Methods

### Experimental setup

**Supercontinuum laser setup.** A supercontinuum laser (NKT Photonics, model EXR-15) was used as the light source to prove the ultrabroadband imaging capability. The spectral bandwidth covered 225 nm from 535 to 760 nm, and the estimated depth resolution was ~0.59 μm. The bandwidth of each wavelength was $\Delta\lambda_S = 15$ $nm$, and the

obtainable depth range was limited to approximately the coherence length 8.88 μm.

**Swept-source laser setup.** For facilitating the volumetric reflection matrix of wider depth range, we adopted a swept-source laser (Superlum BS-840-1-HP, spectral range: 803–878 nm) which emits an approximately monochromatic wave. The bandwidth of each wavelength is so narrow that the depth range can be as large as 5 mm (which corresponds to a spectral linewidth of 0.06 nm). With the swept-source laser, the depth range is rather limited by loss of image information due to defocusing. For achieving a wider depth range, we limited the angular scanning aperture of the objective lens down to 0.4 NA. The estimated depth resolution was ~3.0 μm based on the spectral range of the laser.

## Data availability

The datasets acquired for this study are available from the corresponding authors upon request. The size of the datasets is too large to upload on the publicly accessible repository.

## Code availability

The MATLAB codes used in this work are available from the corresponding author upon request.

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

## Acknowledgements

This work was supported by the Institute for Basic Science (IBS-R023-D1) (Y.-R.L., D.-Y.K.,Y.J., and W.C.), the National Research Foundation of Korea (NRF) grant funded by the Korea government (MSIT) (NRF-2021R1C1C2008158 (Y.-R.L.), NRF-2019R1C1C1008175 (M.K.), and NRF-2021R1A4A5028966 (M.K.)), and the POSCO Science Fellowship of POSCO TJ Park Foundation (Y.-R.L.).

## Author contributions

W.C., Y.-R.L., and D.-Y.K. conceived the experiment, and D.-Y.K., Y.-R.L., Y.J., and M.K. performed experiments. Y.-R.L. and D.-Y.K. analyzed the experimental data. Y.-R.L. and W.C. developed the theoretical framework. Y.-R.L., D.-Y.K., and W.C. prepared the manuscript. All authors contributed to finalizing the manuscript.

## Competing interests

The authors declare no competing interests.
