## [Peer Review File · Nature Communications]

Exploiting volumetric wave correlation for enhanced depth imaging in scattering mediumEditorial Note: This manuscript has been previously reviewed at another journal that is not operating a transparent peer review scheme. This document only contains reviewer comments and rebuttal letters for versions considered at *Nature Communications*.

REVIEWER COMMENTS

Reviewer #1 (Remarks to the Author):

I am aware that I am starting to sound like a broken record, but the problem I highlighted in my first review is still there, largely unresolved: the paper is not clear, and it is unnecessarily hard to read. The new paragraph the authors added does not really add much clarity, and anyway it appears on page 3 of the paper, thus requiring any reader to wade through an enormous amount of text before arriving there. The paragraph that actually explains what the point of the paper is starts on page 5 (line 112). It is a good paragraph, but it should appear much earlier. All the claims made before that paragraph are void of context, and thus pointless.

I (once again) strongly suggest that the authors rethink the way the paper is structured in order to make it clear and readable.

A minor point: the authors have added the information about the mean free path (which is good), but for anisotropic scattering like the one they are discussing here the relevant number is the transport mean free path, which can be significantly larger.

Reviewer #4 (Remarks to the Author):

Review for NCOMMS-22-31219-T “Exploiting volumetric wave correlation for enhanced depth imaging in scattering medium” by Y.-R. Lee et al.

In this paper, a non-invasive volumetric deep optical imaging method based on separating single-scattered light from multiple-scattered light and correction of single-scattered light’s spectro-angular dispersions is demonstrated. Both the idea and the demonstration of this proposed technique are excellent. Although the manuscript is still hard to read after the revision based on the constructive comments of the previous reviewers, I think this version is sufficiently explaining the technique for the experts to reproduce the results and gives the main idea and capability of the technique for the broad audience of Nature Communications.

The authors did a good job in replying to the reviewer comments which improved the readability, and the clarity of main ideas is the method.

In this version, it is clear what can be done with this technique. However, one important missing point is the discussion of the limitations of the technique. In the current version, it gives the

impression that we can image any object buried inside any complex medium at any depth. Therefore, the following comments should be addressed before the manuscript is ready for publication in Nature Communications.

1 – What are the criteria for the object to be imaged buried inside the scattering medium? For example, could one image a certain region of the scattering medium (few TiO₂ particles) buried in the rest of the TiO₂ particles? A discussion on the criteria of the object to be imaged inside the scattering medium must be given in the manuscript.

2 – Can this technique work in a different scattering medium? One missing information in the manuscript is the transport mean free path. Since the TiO₂ particles are large compared to wavelength of light, I expect the scattering to be anisotropic, therefore a much larger transport means free path compared to the scattering mean free path. Can this technique work at depths of 10 transport mean free paths?

3 – It is necessary to show the depth limitation of this technique. Could you show an experimental result at such a depth that this technique fails to work? It is extremely important to show the limitation of this method for the clarity and completeness of this work.

4 – Could this method be used for measuring deposition matrix to deliver optical energy at a target depth similar in reference [N. Bender, et al. Nat. Phys. 18, 309–315 (2022).], but in a three-dimensional medium?

5 – Would there be an advantage of this method for separating single and multiple scattering waves in remission geometry instead of reflection geometry as in reference [N. Bender, et al. PNAS 119, e2207089119 (2022).]?

Reviewer #1

I am aware that I am starting to sound like a broken record, but the problem I highlighted in my first review is still there, largely unresolved: the paper is not clear, and it is unnecessarily hard to read. The new paragraph the authors added does not really add much clarity, and anyway it appears on page 3 of the paper, thus requiring any reader to wade through an enormous amount of text before arriving there. The paragraph that actually explains what the point of the paper is starts on page 5 (line 112). It is a good paragraph, but it should appear much earlier. All the claims made before that paragraph are void of context, and thus pointless.

I (once again) strongly suggest that the authors rethink the way the paper is structured in order to make it clear and readable.

We believe that our manuscript provides enough clarity for readers in imaging science to grasp the main findings. In the previous round, we added a figure (Fig. 1b) and a paragraph on page 5 to introduce the basic background of the problems occurring in imaging within complex scattering medium following the reviewer's suggestion. Since the reviewer seems to find this addition helpful, we added the essential part of the paragraph on page 5 to the early part of the introduction. This will help readers to find the difficulty in deep imaging from the beginning. Below are the revised texts.

“Complex media such as brain tissues under a skull induces both multiple scattering and sample-induced wave retardations, which make it difficult to extract single-scattered waves carrying the object information for high-resolution imaging. The single-scattered component of each planar wave constituting a focused illumination is attenuated exponentially with depth due to multiple scattering. The wave retardations further attenuate the peak intensity by hampering the constructive interference of the single-scattered waves in forming a focused spot.”

A minor point: the authors have added the information about the mean free path (which is good), but for anisotropic scattering like the one they are discussing here the relevant number is the transport mean free path, which can be significantly larger.

Based on the Mie solution, the anisotropic factors of the polystyrenes and TiO₂ particles with a mean diameter of 1 μm in PDMS are estimated to be 0.91 and 0.38, respectively. Therefore, their respective transport mean free paths are $l_t = 11 l_s$ and $1.6 l_s$. We specified these numbers in the revised manuscript.

We would like to emphasize that the scattering mean free path is a more relevant parameter for evaluating the sample in our study because the proposed method relies on the volumetric wave correlation of ballistic components for reconstructing high-resolution images. The transport mean free path is more relevant for wavefront shaping methods where the energy delivery enhancement by controlling multiple scattering, not the high-resolution imaging, is a major concern.

Reviewer #4

Review for NCOMMS-22-31219-T “Exploiting volumetric wave correlation for enhanced depth imaging in scattering medium” by Y.-R. Lee et al.

In this paper, a non-invasive volumetric deep optical imaging method based on separating single-scattered light from multiple-scattered light and correction of single-scattered light's spectro-angular dispersions is demonstrated. Both the idea and the demonstration of this proposed technique are excellent. Although the manuscript is still hard to read after the revision based on the constructive comments of the previous reviewers, I think this version is sufficiently explaining the technique for the experts to reproduce the results and gives the main idea and capability of the technique for the broad audience of Nature Communications.

The authors did a good job in replying to the reviewer comments which improved the readability, and the clarity of main ideas is the method.

We deeply appreciate the reviewer's taking the time to carefully read our manuscript and acknowledging the conceptual novelty and capability of the proposed ideas and the clarity of the revised manuscripts.

In this version, it is clear what can be done with this technique. However, one important missing point is the discussion of the limitations of the technique. In the current version, it gives the impression that we can image any object buried inside any complex medium at any depth. Therefore, the following comments should be addressed before the manuscript is ready for publication in Nature Communications.

We thank the reviewer's thoughtful comments. In the following, we discussed the limitations of the technique in detail.

1 – What are the criteria for the object to be imaged buried inside the scattering medium? For example, could one image a certain region of the scattering medium (few TiO₂ particles) buried in the rest of the TiO₂ particles? A discussion on the criteria of the object to be imaged inside the scattering medium must be given in the manuscript.

In fact, the reconstructed images in Figs. 6i-k were TiO₂ particles in a certain region of the scattering medium buried in the rest of the TiO₂ particles.

As the reviewer commented, quantitative criteria exist for the proposed method to work. Essentially, a single scattering signal from the target depth competes with multiple scattering backgrounds from all the other depths. Therefore, their relative strength sets one important criterion. There is no specific requirement for the shape, size, and refractive index of the target objects as long as they produce sufficient backscattering signals. The complexity of the scattering medium in terms of inducing spectro-angular dispersions is another defining parameter. Finally, the number of orthogonal channels covered by the volumetric reflection matrix sets the working condition.

In the main text on page 14, we provided the quantitative criteria of the working condition. Our method can work when $C_{\text{rel}} \approx \frac{S}{M} \zeta \sqrt{N}$, which defines the relative contribution of single- to multiple-scattered waves to the correlation, is greater than a certain threshold (C_{th}), the specific value of which depends on noise level (see Supplementary section VII for full derivation and its experimental validation). Here, S and M are the average intensities of single- and multiple-scattered waves at each detection channel, respectively. N is interpreted as the total number of elements involved in the cross-correlation of pupil

functions. ζ represents the complexity of both input and output angular dispersions, and $\zeta = 1$ when both input and output dispersions do not exist. Here, S is determined by the reflectance of the target object, which is determined jointly by the sizes, shapes, and refractive index contrasts of the objects located at the focal plane, and the depth of the object. M and ζ are determined by the properties of the scattering medium.

Following the reviewer's suggestion, we added the following discussion to the discussion section of the revised manuscript.

“The working condition of the proposed approaches is set by the parameter $C_{\text{rel}} = \frac{S}{M} \zeta \sqrt{N}$. For our method to work, the single-scattering intensity S set by the reflectance and the depth of the target object, multiple scattering intensity M , and the complexity of the spectro-angular dispersion ζ should meet the criteria such that C_{rel} should be larger than a certain threshold.”

2 – Can this technique work in a different scattering medium? One missing information in the manuscript is the transport mean free path. Since the TiO₂ particles are large compared to wavelength of light, I expect the scattering to be anisotropic, therefore a much larger transport means free path compared to the scattering mean free path. Can this technique work at depths of 10 transport mean free paths?

Based on the Mie solution, the anisotropic factors of the polystyrenes and TiO₂ particles with a mean diameter of 1 μm in PDMS are estimated to be 0.91 and 0.38, respectively. Therefore, their respective transport mean free paths are $l_t = 11 l_s$ and $1.6 l_s$. We specified these numbers in the revised manuscript.

We would like to emphasize that the scattering mean free path is a more relevant parameter for evaluating the sample in our study because the proposed method relies on the volumetric wave correlation of ballistic components for reconstructing near-diffraction-limited high-resolution images. The transport mean free path is more relevant for wavefront shaping methods where the energy delivery enhancement by controlling multiple scattering, not the high-resolution imaging, is a major concern. For this reason, we specified the imaging depth of our methods in terms of the scattering mean free path, which is around 10-12 l_s .

3 – It is necessary to show the depth limitation of this technique. Could you show an experimental result at such a depth that this technique fails to work? It is extremely important to show the limitation of this method for the clarity and completeness of this work.

As explained in response to comment #1, our method can work when $C_{\text{rel}} \approx (S/M) \zeta \sqrt{N} \geq C_{\text{th}}$. Since S/M changes exponentially with depth, it is difficult to find the exact depth limit by changing the depth of the target. Therefore, we examined the working depth by changing the number of input angles (N_θ) and the number of wavelengths (N_λ) in Supplementary section VII. We examined this condition after coherence gating. Therefore, S/M depends linearly on the bandwidth, or the number of wavelength channels, N_λ . N is proportional to the number of input angular channels N_θ for a given number of output channels. Essentially, C_{rel} is proportional to $N_\lambda \sqrt{N_\theta}$ such that the depth limit is set by $N_\lambda \sqrt{N_\theta} = \text{constant}$.

Figure R1 shows the image reconstruction results for various combinations of bandwidth (N_λ) and number of input channels (N_θ) with the sample used in Fig. 4 (the effective optical thickness of $9.4 l_s$). The conditions where the image reconstruction is successful are indicated in the green-shaded area,

while those where the image reconstruction failed were indicated in the red-shaded area. The borderline between the two shaded areas approximately matches the contour set by $N_\lambda \sqrt{N_\theta} = \text{constant}$, as expected.

When the bandwidth is 225 nm, the method works until N_θ decreases to 600 in Fig. R1. The total number of independent input angular channels of the system is approximately 5500 (NA=1.0 and field of view=22.8 $\mu\text{m} \times 22.8 \mu\text{m}$ at 542.5 nm). If we use the full number of input channels, the method will work even when S/M is further attenuated by $\sqrt{600/5500}$.

The imaging depth limit of this sample can be deduced from the attenuation of single- and multiple-scattered waves depending on imaging depth. In the reflection-mode imaging, the intensity of single-scattered waves S and that of multiple-scattered waves M can be described as $S(z) = e^{-2z/l_s}$, and $M(z) = e^{-2z/l_m}$, respectively. Here l_s corresponds to the mean free path of single-scattered waves, and l_m represents the attenuation length of multiple scattering. l_m depends on various factors such as the numerical aperture, field of view, and the types of gating operations. It is longer than l_s in general, e.g. typically, $l_m \sim 1.5 l_s$ when the temporal gating is applied. The SMR ($\equiv S/M$) can be described as $\text{SMR}_0(z) = e^{-\frac{2z}{l'}}$, where $l' = \frac{l_m l_s}{l_m - l_s}$ corresponds to the attenuation length of SMR. Let us set the original imaging depth $z_0 (= 9.4 l_s)$ as $C_{\text{rel}}(z_0) \approx \text{SMR}_0(z_0) \zeta \sqrt{600} = C_{\text{th}}$. If we increase N_θ to 5500, the achievable imaging depth z_d is given by the condition, $C_{\text{rel}}(z_d) \approx \text{SMR}_0(z_d) \zeta \sqrt{5500} = C_{\text{th}}$.

Therefore, the imaging depth limit of this sample is estimated to be

$$z_d = z_0 + \frac{l_m l_s}{l_m - l_s} \frac{\ln(\sqrt{5500/600})}{2} \sim 9.4 l_s + 1.7 l_s = 11.1 l_s.$$

We added this discussion to Supplementary section VII.2 of the revised manuscript.

Figure R1. Image reconstruction results of VRM depending on N_λ and N_θ . The data in Fig. 4 of the manuscript was analyzed for various bandwidths (or the number of wavelength channels N_λ) and the number of input angular channels N_θ . The green box indicates a successful reconstruction, the red box indicates a failed reconstruction, and the yellow box indicates a borderline image. Successful

reconstructions are shaded in green, while failed reconstructions are shaded in red.

4 – *Could this method be used for measuring deposition matrix to deliver optical energy at a target depth similar in reference [N. Bender, et al. Nat. Phys. 18, 309–315 (2022).], but in a three-dimensional medium?*

That is a very interesting question. Measuring the deposition matrix at a 2D medium has been possible, as shown in the reference paper mentioned by the reviewer. However, measuring the deposition matrix in a three-dimensional medium is impossible unless a detector is located inside the medium. Studies employing the acousto-optic interaction resemble the most with the deposition matrix recording as the ultrasound focus sets the detection volume inside the 3D scattering medium (Katz, O., Ramaz, F., Gigan, S. & Fink, M. *Nat. Commun.* **10**, 717 (2019); Jang, M. *et al. Nat. Commun.* **11**, 710 (2020)). Our method exploiting all the incident angles and wavelengths can be applied to measure the deposition matrix of the 2D medium and acousto-optic interaction in the 3D medium for the optimal enhancement of light energy delivery at the specific depth. We added this interesting reference suggested by the reviewer to Ref. [31] of the revised manuscript.

The use of spatio-spectral information can also offer new avenues for studying mesoscopic wave correlations and their use for wave focusing and light energy delivery^{25–27,30,31,32}.

5 – *Would there be an advantage of this method for separating single and multiple scattering waves in remission geometry instead of reflection geometry as in reference [N. Bender, et al. PNAS 119, e2207089119 (2022).]?*

We think that the potential benefit of our approach to the interesting study of enhancing light energy delivery in remission geometry is to allow the use of both spatial and spectral degrees of freedom for wavefront shaping. This can potentially increase the degree of light energy delivery enhancement. The separation of single- and multiple-scattered waves will be less interesting in the remission geometry because the waves traveling through the banana region are mostly multiple-scattered waves. We added this interesting reference paper to Ref. [32] of the revised manuscript.

The use of spatio-spectral information can also offer new avenues for studying mesoscopic wave correlations and their use for wave focusing and light energy delivery^{25–27,30,31,32}.

REVIEWERS' COMMENTS

Reviewer #4 (Remarks to the Author):

All my comments are addressed clearly in the new version. I especially appreciate the detailed analysis on the dependence of the reconstruction fidelity of the images on the bandwidth and number of input channels in the supplementary document. Therefore, I recommend the current version of this manuscript to be published in Nature Communications.